# Calcium dynamics in habenular astrocytes regulate active coping within behavioral transitions
Léo Michel [1], Denys Osypenko[1], Patricia Molina[1], Kadir A. Mutlu [2], Salvatore Lecca [1], Chihiro Hisatsune [3], Katsuhiko Mikoshiba [4], Toko Kikuchi [5], Emre Yaksi [2,6], Andrea Volterra[1,5] & Manuel Mameli [1,7] ✉

Behavioral challenges prompt alternating vigorous and reduced mobility – active, passive coping – that optimize energy investment. Here, we show that disrupting astrocytes calcium signaling in the mouse lateral habenula (LHb) prolongs active coping. This state manifests through calcium elevations in both mouse and zebrafish habenular astrocytes. Presynaptic tracing approaches integrate LHb astrocytes within aversion–related neuronal circuits. Thus, astrocytes regulate state transitions highlighting their computational contribution to behaviors across species.

When actions in response to threats fail to ensure safety, animals often suppress them and become passive. This behavioral transition allows conserving energy between escape attempts or enabling alternative defensive methods[1,2]. Accordingly, in both rodents and zebrafish, threats and discomfort drive a rapid alternation between active and passive states whereby an initial vigorous mobility (active coping, AC) eventually transitions into a passive coping (PC) state of quiescent activity[1,3–6]. Notably, when inescapable adversities are excessive, ACs become futile across time ultimately leading to helplessness (extensive PC) – a core clinical feature of major depression[1,6]. Thus, understanding the optimal weight between AC and PC is paramount to decipher aspects of animal behaviors and human mental disorders. Here we examined the role of astrocytes in the lateral habenula (LHb) – a core aversion–encoding brain center[7] – in shaping coping behaviors.

## Results

### Habenular astrocytes regulate behavioral state transitions

We employed a behavioral challenge protocol whereby C57Bl6j mice experienced an inescapable condition across time. Mice were first habituated to a head–fixation frame and experimental settings. Forelimbs and body movements were then video tracked (based on pixel movement detection), automatically detected, and quantified throughout (Supplementary Video 1; Fig. 1a; Supplementary Fig. 1a, b). During the six–minute inescapable sessions, mice exhibited AC bouts that alternated with PC (Fig. 1a; Supplementary Fig. 1a, b). The duration and magnitude of AC and

PC remained stable over time, indicating that behavioral transition is a stable and intrinsic coping process in mice (Fig. 1b, c).

External behavioral challenges (threats) in freely behaving mice drive escape or avoidance[8]. We examined whether these stimuli delivered to head-fixed mice are equally capable to generate AC–to–PC transitions. Mice were successively exposed to one of three external behavioral challenges: airpuffs (0.5 s at 0.5 bar), aversive sounds (17–20 kHz and 80 dB), and water drops (15 repetitions/stimulus over three days; Supplementary Fig. 1c–o). When external behavioral challenges were introduced in the task, all these modalities led to instances of time–locked AC bouts followed by PC (Supplementary Fig. 1c–o). The probability of triggering AC bouts throughout the session remained 20–50% across the different behavioral challenges (Supplementary Fig. 1e, j, o). Notably, the stability of spontaneous AC to PC transitions remained largely unaffected by the presence of external behavioral challenges (Supplementary Fig. 1f–g, k–l, p–q). Altogether, these findings reveal that AC and behavioral transitions occur spontaneously, remain stable across time, and can be driven by external challenges according to their nature or intensities.

Astrocytes are brain cells that integrate behavioral states, orchestrate neural circuitries and thereby affect behaviors[9–12]. For instance, while calcium signaling in zebrafish astrocytes optimizes AC–to–PC transitions, their dynamics in the cortex modulate depressive–like behaviours[4,5,13,14]. Additionally, astrocytes in the LHb contribute to stress responses and negative affective states in rodent models of depression, a state featuring extensive PC[15–17]. Thus, we examined habenular astrocytes contribution to

[1]The Department of Fundamental Neuroscience, The University of Lausanne, Lausanne, Switzerland. [2]Kavli Institute for Systems Neuroscience and Center for Algorithms in the Cortex, Norwegian University of Science and Technology, Trondheim, Norway. [3]Calpain Project, Tokyo Metropolitan Institute of Medical Science, Tokyo, Japan. [4]Shanghai Institute for Advanced Immunochemical Studies, ShanghaiTech University, Shanghai, China. [5]Wyss Center for Bio and Neuro Engineering, Campus Biotech, Geneva, Switzerland. [6]Koç University Research Center for Translational Medicine, Koç University School of Medicine, Istanbul, Turkey. [7]Inserm, UMR-S 839, Paris, France. ✉e-mail: manuel.mameli@unil.ch

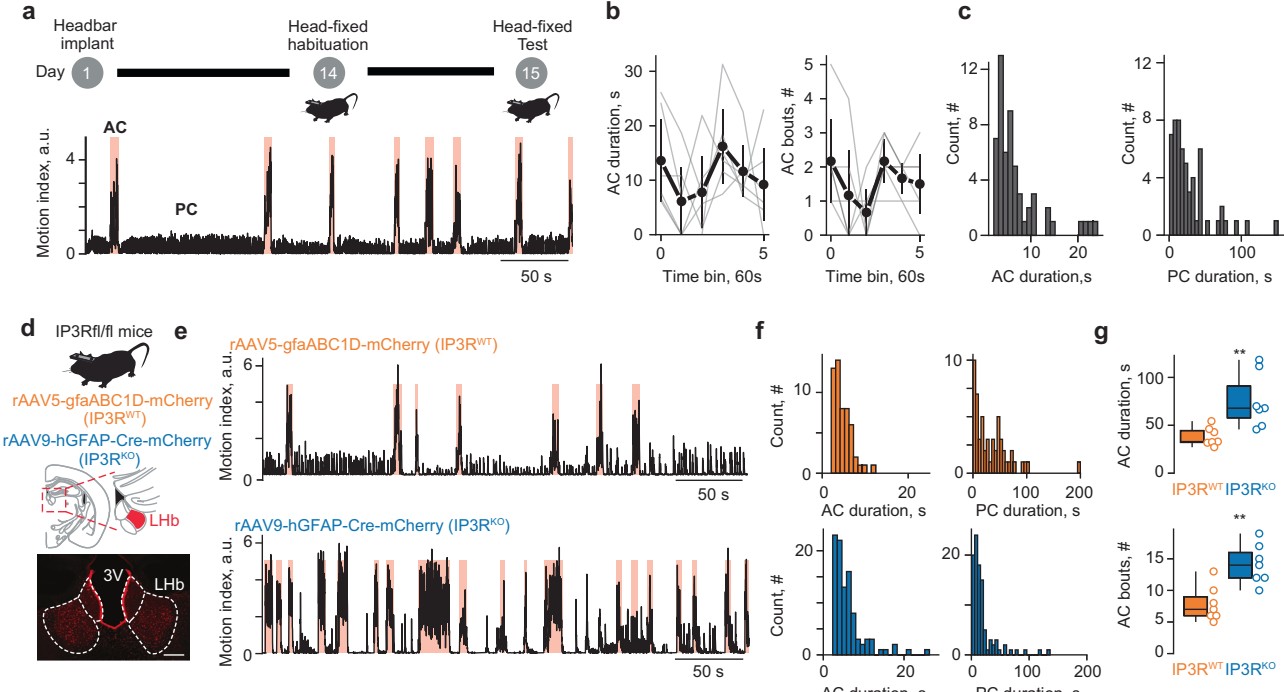

**Fig. 1 | Habenular astrocytes temporally regulate active–to–passive behavioral state transitions in mice. a** Top: schematic of experimental design. Bottom: sample trace of motion index during active–passive behavioral state, light–red squares show active coping (AC) throughout the task. Passive coping (PC). **b** Left: Time binned population-average AC duration throughout the task ($n = 6$ mice; bin=1 min), ANOVA $F_{5, 25} = 1.41$ $p = 0.29$. Right: same as left but for AC bout counts, ANOVA $F_{5, 25} = 2.06$ $p = 0.19$. **c** Left, Histogram reflecting the duration of all counted AC bouts ($n = 6$ mice). Right, same as left but for PC bouts. **d** Top: schematic of experimental strategy. Bottom: representative image of rAAV9-hGFAP-Cre-mCherry bilateral LHb expression in IP3R[KO] mice; scale bar, 200 μm. **e** Sample trace

of AC to PC transitions in a IP3R[fl/fl] mouse virally injected with a control rAAV5-gfaABC1D-mCherry (IP3R[WT], top), or a rAAV9-hGFAP-Cre-mCherry virus (IP3R[KO], bottom). Light–red squares indicate AC. **f** Histogram of all counted AC bouts duration (left) and PC bouts duration (right) in IP3R[WT] (top, orange) and IP3R[KO] (bottom, blue) ($n = 7$ mice/group). **g** Total AC duration (top) and bouts number (bottom). Data are presented as box plot, error bars indicate min to max, median and scatter in orange for IP3R[WT] and blue for IP3R[KO] mice. Two-sided unpaired t-test, $t_{12} = 3.279$ **$p = 0.007$ for total AC duration and $t_{12} = 3.962$ **$p = 0.002$ for number of AC bouts.

AC and PC states in mice, by specifically disrupting calcium signaling in this cell–type. Astrocytic calcium signaling largely relies on calcium rise from the endoplasmic reticulum (ER) via the inositol 1,4,5-trisphosphate receptors (IP₃Rs)[18]. We employed a transgenic mouse model enabling Cre-dependent *Itpr1,2,3* (IP3R1,2,3 subtypes, IP3R[fl/fl]) genes deletion in LHb astrocytes (IP3R[KO], Fig. 1d). Five weeks after LHb bilateral injection of a rAAV9-hGFAP-Cre-mCherry, the number of AC bouts and total AC time increased in IP3R[KO] head-fixed mice compared to control virus–injected littermates (IP3R[WT]; Fig. 1e–g; Supplementary Fig. 2a–d). Thus, disrupting LHb astrocytes calcium signaling expands AC at the expense of PC. We then virally expressed a 122-residue inhibitory peptide from β-adrenergic receptor kinase1 (rAAV5-GfaABC1D-iβARK-p2a-mCherry) in LHb astrocytes – a construct capable of wiping out astrocytic calcium elevations throughout the brain (Supplementary Fig. 2e)[17,19]. Similarly to IP3R[KO] mice, iβARK overexpression expanded AC (Supplementary Fig. 2f–h). We then set out to emulate Ca²⁺ increases in astrocytes. We expressed a designer receptor exclusively activated by designer drugs (hM3D(Gq); Supplementary Fig. 2i), and used chemogenetic stimulation by the designer drug clozapine *N*-oxide (CNO, i.p., 1 mg/kg). Alternated CNO and saline injections led to AC bouts fluctuations that were opposite to those in IP3R[KO] or iβARK overexpression (Supplementary Fig. 2j–l). Altogether, this suggests that astrocytes in the LHb temporally constrains AC and affects the balance between AC and PC when facing behavioral challenges.

### Astrocyte dynamics and their input organization
To examine LHb astrocytes activity during AC, we recorded their calcium signaling dynamics through fiber photometry in adult mice. We virally expressed the genetically encoded calcium indicator GCaMP (versions 6 f

and 8 s) under the astrocyte–specific promoter GfaABC1D[20] (Fig. 2a, b; Supplementary Fig. 3a, b). When mice were head–fixed, we observed spontaneous AC bouts occurring along with robust increase in LHb astrocytes calcium fluorescence, which returned to baseline during PC bouts (Fig. 2b–d; Supplementary Fig. 3c, d). The rise and amplitude of calcium transients positively correlated with the length of AC supporting their relationship (Fig. 2d; Supplementary Fig. 3e). Notably, astrocytic calcium elevations were specific to AC. Using the tail suspension as a challenging experience promoting AC to PC transitions, led to AC–aligned increase in LHb astrocytes calcium fluorescence (Supplementary Fig. 3f–i). Next, we observed that mice freely moving in an open field did not show astrocytes dynamics time–locked with bouts of increased velocity as opposed to AC during head fixation (Supplementary Fig. 3j–s). In support of this observation, complementary motor actions in freely moving animals (grooming and digging), social interactions or palatable rewards failed to produce time–locked transients (Supplementary Fig. 3t–x). Behavioral challenges can produce time–locked AC bouts followed by PC (Supplementary Fig. 1c–o). Thus, we sought to determine whether LHb astrocytes universally encode AC independently of the stimulus nature that triggers it (intrinsic or externals). When exposing mice to external behavioral challenges, calcium responses emerged aligned to the onset of airpuffs, aversive sounds, water drops as well as aversive light and footshocks (Supplementary Fig. 4a–h). These challenges reliably increased astrocytic calcium transients although with different amplitudes and temporal dynamics. This might emerge from variance in the number of GCaMP-expressing astrocytes or relate to the negative value carried by each challenge potentially explaining their suboptimal and variable capacity of eliciting AC. Mechanistically, AC–mediated astrocytic calcium dynamics may emerge from

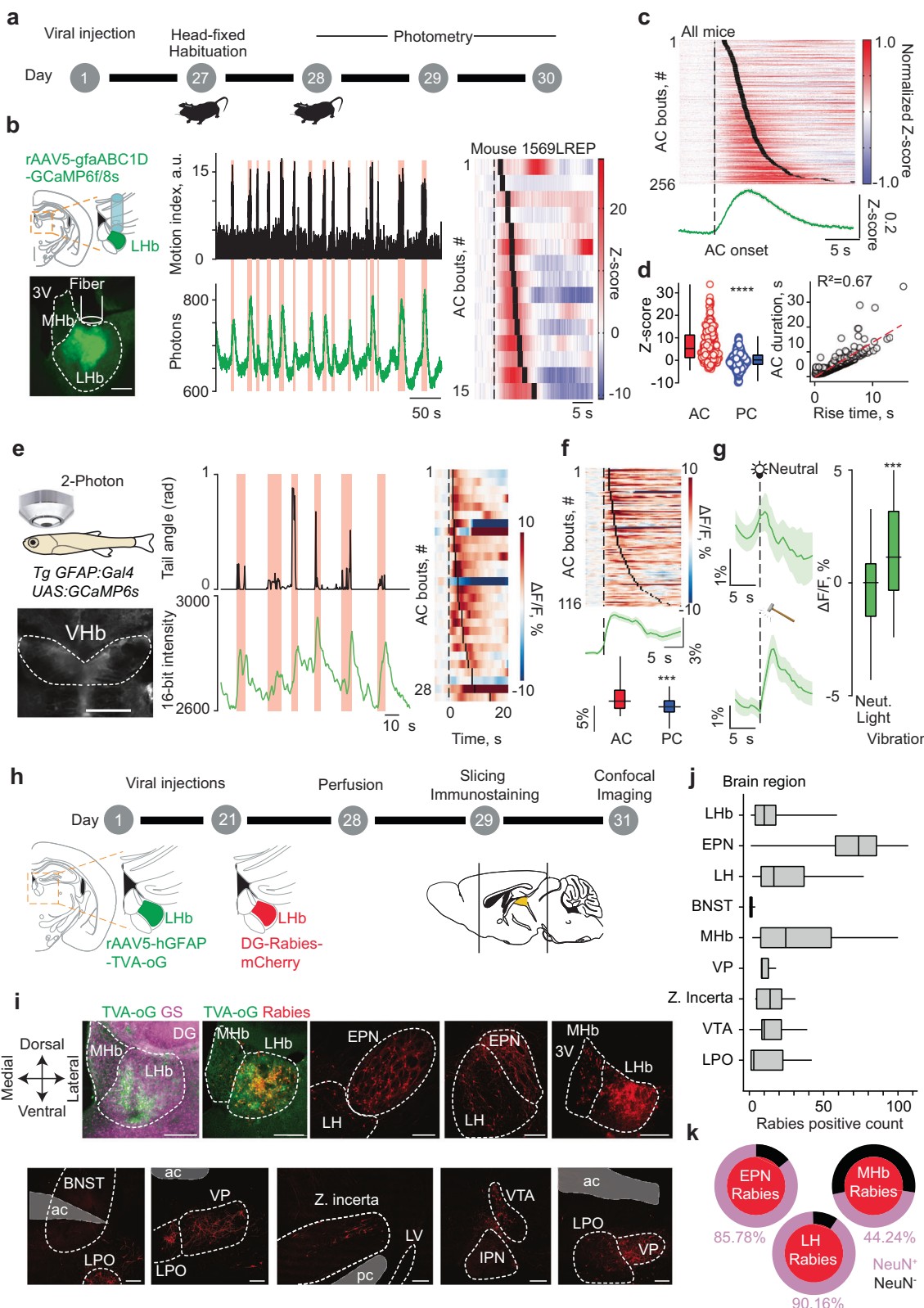

neuron–astrocyte crosstalk[17,19,21,22]. Expression of GCaMP8s in astrocytes of the LHb within one hemisphere and the red fluorescent calcium sensor jRGECO in neurons of the contralateral LHb revealed AC–locked and short latency neuronal excitation occurring along with astrocytic dynamics (Supplementary Fig. 4i–k). Latency analysis indicated that astrocytic calcium elevations were preceded by AC–driven neuronal transients

(Supplementary Fig. 4l, m)[17]. Altogether, this corroborates the notion that a tight coupling between neurons and astrocytes across different behavioral states exists in the LHb[16,17].

The computation of AC and PC in response to threats is also a feature of zebrafish astrocytes[1,4,5]. To examine if habenular astrocyte recruitment during AC is conserved across species, we employed a head-restrained and

**Fig. 2 | LHb Astrocyte dynamics during active coping states and input organization. a** Schematic of experimental design for photometry experiments. **b** Left panel, schematic depicting LHb viral injection and optic fiber placement (top) and representative image of rAAV9-GfaABC1D-GCaMP8s expression and fiber track in LHb (bottom). 3 V: Third ventricle; MHb: medial habenula. Middle panel: Sample motion index trace from single mouse and aligned raw photometric signal in green. Light-red squares indicate AC. Right panel: Sample heatmap related to photometry signal across trials. Data are presented as a heatmap of Z-score for each AC bout, sorted from shortest to longest, dashed line shows onset of AC while thick black lines show offset. **c** Top: heatmap of normalized Z-score (per animal) for each AC bout ($n = 14$ animals, 256 trials), sorted and represented as (**b**). Bottom: aligned averaged and normalized Z-score trace for all AC bouts. **d** Left: average Z-score for the last second of AC and PC bout. Data are presented as box plot, error bars indicate min to max, median and scatter. Two-tailed Wilcoxon signed rank test, ****$p < 0.0001$. Right: calcium rising time in function of AC bout duration, red dashed line shows regression line, R$^2$ = 0.67 ****$p < 0.0001$. **e** Left panel, Illustration of a head-restrained, awake, behaving juvenile zebrafish under a two-photon microscope (top) and a raw image of a Tg(GFAP:Gal4;UAS;GCaMP6s) zebrafish expressing GCaMP6s in astroglial cells of the ventral habenula (bottom). Middle panel, top: Example trace of zebrafish locomotion measured as tail beat angle. Red shading indicates the duration of locomotor AC bouts. Bottom, raw calcium signal from astroglial cells in the ventral habenula of the zebrafish. Each red line marks the onset of a locomotor AC bout. Right panel, top: Astroglial calcium traces [relative change in calcium signals (ΔF/F)] during locomotor AC bouts from an example fish. Warm

colors represent increased activity, cold colors represent decreased activity. The dashed line indicates AC bout onset, and the thick black line represents AC bout offset. **f** Astroglial calcium traces during locomotor AC bouts from all recorded fish ($n = 5$). Bottom, Comparison of average astroglial activity between active and passive coping periods. **g** Left, Mean light-evoked astroglial calcium signals from all animals after consecutive exposure to neutral light (top, $n = 5$ animals, 30 trials) or aversive vibrations (bottom, $n = 5$ animals, 25 trials). Right, distribution of average light-, and vibration-evoked activity in trials from all animals Wilcoxon signed-rank test, ***$p < 0.001$. Data are presented as box plot, error bars indicate min to max, median and scatter. **h** Schematic of experimental design for retrograde rabies strategy. **i** Representative images depicting LHb injection site for rAAV5-GFAP-TVA-oG counterstained with an anti-Glutamine synthetase (GS) antibody, after ΔG-Rabies-mCherry injections and brain regions with positive rabies positive cells. Scale bar, 200 µm. 3 V, third ventricle; ac anterior commissure, BNST bed nucleus stria terminalis, EPN entopeduncular nucleus, IPN interpeduncular nucleus, LH lateral hypothalamus, LPO lateral preoptic area, LV lateral ventricle, MHb medial habenula; pc posterior commissure, VP ventral pallidum, VTA, ventral tegmental area. **j** Rabies positive brain region cell count. Brain regions were considered when exceeding >5 rabies positive cells ($n = 7$ animals). Data are presented as box plot, error bars indicate min to max, and median. **k** Quantification of neuronal and non-neuronal rabies positive cells in EPN, MHb and LH. Data are presented as pie charts representing RV$^+$NeuN$^+$ cells in magenta and RV$^+$NeuN$^-$ cells in black over total RV$^+$ cells ($n = 7$ animals).

tail–free juvenile zebrafish preparation expressing GCaMP6s under the *GFAP* promoter *Tg(GFAP:Gal4; UAS:GCaMP6s)*[5,23] (Fig. 2e). In zebrafishes, spontaneously occurring locomotor tailbeats (AC), computed through angle variations, elicited time-locked calcium transients in the ventral habenula (vHb, LHb homologue[24] astrocytes (Fig. 2e–f). Similar to mice, average fluorescence signal was larger during AC bouts compared to PC periods in zebrafishes (Fig. 2f, Supplementary Fig. 4n, o). Exposure to consecutive neutral visual stimuli or aversive vibrations[25] unraveled that vHb astrocytes in zebrafish specifically increase their calcium dynamics in response to the latter (Fig. 2g, Supplementary Fig. 4p, q). Thus, transient habenular astrocytic activity is remarkably similar in mice and zebrafish suggesting that astrocytic computations of AC–PC transitions are recurrent across species.

To better understand the circuit mechanisms underlying the control of AC by LHb astrocytes, we used modified rabies–based viral strategies to trace inputs onto this cell population[26] (Fig. 2h). Starter cells expressing rabies were limited to LHb astrocytes as shown by positive immunostaining for glutamine synthetase (GS)[27] but not the neuronal marker NeuN (Fig. 2i; Supplementary Fig. 5a–c). Aversion–encoding structures including the entopeduncular nucleus (EPN), the lateral hypothalamus (LH), the ventral tegmental area (VTA), the ventral pallidum (VP) and the lateral preoptic area (LPO) emerged as containing rabies positive cells (Fig. 2i–j; Supplementary Fig. 4a). This connectivity signature is similar to long–range inputs onto LHb neurons[7,28]. However, unlike neurons in LHb, rabies in astrocytes did not lead to labeling of the bed nucleus of the stria terminalis (BNST)[29,30] (Fig. 2i, j). Furthermore, LHb astrocytes–limited rabies approaches unraveled the medial habenula (MHb), a projection yet not reported for neurons (Fig. 2i, j). Immunostaining with the neuronal marker NeuN revealed that long–range projections from the EPN and LH emerged virtually solely from neurons (Fig. 2k; Supplementary Fig. 5d, e). Instead, rabies positive cells in MHb and LHb were both neurons and astrocytes (Fig. 2k; Supplementary Fig. 5d, e). Accordingly, rAAV1-hSyn1-mCherry injections targeting neurons in the MHb and concomitant EGFP expression in LHb astrocytes unraveled MHb axons within the LHb glial territory (Supplementary Fig. 5f). Altogether, these data indicate that LHb astrocytes may compute AC–related signals, at least partly, through local connectivity within the habenular complex as well as limbic and aversion–recruited neuronal networks.

## Discussion
The LHb and its neuronal populations have long been associated with aversive encoding that supports both physiological and pathological

behaviors[7]. Here, we demonstrate that LHb astrocytes of mice and zebrafish function as computational elements for AC emerging after inescapable experiences (i.e. during head fixation and behavioral challenges). Our findings expand the role of astrocytes to include the integration of state and behaviorally–relevant information[31–35], enabling a critical balance between effective and futile actions, thereby regulating investment–driven expenditures[4,13].

This study establishes astrocytes as a fundamental module within the habenula, orchestrating adaptive behaviors and modulating the AC state in mouse and zebrafish. This aligns with observations whereby LHb astrocytes contribute to depressive–like behavioral features hallmarked by extended passive coping[15,17]. Hence, our data underscore the contribution of habenular astrocytes to behavioral states in physiological contexts and highlight the cross–species conservation of astrocytic involvement during AC to PC transitions[1,4,5,7].

Mechanistically, astrocytes across species (Drosophila, zebrafish, and rodents) influence nearby neurons through gliotransmission, whereby glutamate, d-serine, and GABA glial release modulate neuronal activity and synaptic transmission[36–39]. In the LHb, astrocytes regulate neuronal properties through release of the gliotransmitter ATP/adenosine as well as titrating extracellular potassium, release of noradrenaline and endocannabinoid signaling[16,40]. Recent findings described a stress–induced recruitment of LHb neurons, norepinephrine, and astrocytes[17]. LHb neurons excitation leads to increased astrocytic calcium surge via α1A-AR-dependent norepinephrine signaling. This in turn, promote a gliotransmitter (glutamate and ATP/adenosine) cascade that further extends LHb neural activity. Thus, LHb astrocytes may use these glia–neuron interactions to define the extent of AC and ultimately impact AC–PC balance. Behavioral states synchronize astrocytic calcium with the local vascular system[41]. Future studies may bridge these insights across AC–PC transitions to determine the energy demand required for it.

Structurally, astrocytic processes enwrap synapses in the brain[36]. Our study identified a subcortical network projecting to LHb astrocytes, with EPN and LH emerging as structures supporting astrocytic and neuronal contributions to negative affect[42,43]. While these inputs impinge on both LHb neurons and astrocytes, notable specificities emerged. For instance, MHb neurons, that appear not to project to LHb neurons, may instead functionally engage LHb astrocytes[28]. Our labeling strategy also revealed LHb and MHb astrocytes expressing rabies but not the TVA receptor, suggesting either active retrograde rabies transfer or potential astrocytes released viral–containing micro vesicles[36]. In contrast, BNST projections, which

innervate LHb neurons, do not emerge as astrocyte inputs[30,42]. Altogether, these findings support that specialized astrocytes exist within specific circuitries of the central nervous system[17,44].

Thus, non–neuronal cell types orchestrate energy–saving behaviors across species offering a neural circuit framework for the optimization of fitness and survival.

# Methods
## Animals
All procedures were done in accordance with the veterinary office of Vaud (Switzerland, license VD3798c). Animals (Males and females C57BL6/6JR, Janvier, 8–20 week–old mice (*mus musculus*) and *Itpr1,2,3fl/fl* mice, BRC No. RBRC10292 provided by RIKEN BRC.) were maintained on a 12 hr/12 hr light/dark cycle in individually ventilated cages enriched with nesting material, tubes (IVC, Innovive, France) and were fed ad libitum. A protocol including the research question and experimental design was not registered prior the study. Behavioral experiments were carried out during the light cycle and no humane endpoints were included in the study.

**Fish husbandry**. NFSA (Norwegian Food Safety Authority) has approved the animal facility and fish maintenance. We have complied with all relevant ethical regulations for animal use. Fish (*Danio Rerio*) were kept in 3,5 liters tanks in a Tecniplast ZebTec Multilinking System. Constant conditions were maintained: 28.5 °C, pH 7.2, 700 μSiemens. 14:10 hour light/dark cycle was preserved. Juvenile (3 to 4-week-old) zebrafish Tg(GFAP:Gal4;UAS;GCaMP6s) was used to image astroglia calcium signals. Dry food (SDS100 up to 14dpf and SDS 400 for adult animals, Tecnilab BMI) was given to fish twice a day, in addition to Artemia nauplii (Grade 0, Platinum Label, Argent Laboratories, Redmond, USA) once a day. From fertilization to 3dpf (days post fertilization) larvae were kept in a Petri dish with egg water (1.2 g marine salt in 20 L RO water, 1:1000 0.1% methylene blue) and between 3 and 5dpf in artificial fish water (AFW: 1.2 g marine salt in 20 L RO water). All experimental procedures performed on zebrafish larvae and juveniles were in accordance with the Directive 2010/63/EU of the European Parliament and the Council of the European Union and approved by the Norwegian Food Safety Authorities.

## Surgical procedures
Animals were anesthetized with an i.p. injection of ketamine (150 mg/kg) and xylazine (100 mg/kg) and were placed on a stereotactic frame (Kopf, Germany). The ocular protector Viscotear was used to prevent eye damage. Surgery was performed on a heating pad to keep a stable body temperature. The scalp was opened, and holes were drilled above the LHb (in mm AP −1.45, ML 0.45, DV from brain surface 2.6). AAV infusions (rAAV5-GfaABC1D-GCaMP6f; rAAV9- GfaABC1D-GCaMP8s; rAAV5-hsyn-jRGECO1a; rAAV9-hGFAP-Cre-mCherry; rAAV5-hGFAP-TVA-oG-EGFP; ΔG-Rabies-mCherry, Zurich Viral Vector Core; rAAV5-GfaABC1D-iβARK-p2A-mCherry; rAAV9-hGFAP-hM3D(Gq)-mCherry) were performed through a glass needle at a rate of 100 nl min$^{-1}$. The injection pipette was withdrawn from the brain 10 min after the infusion. When required, a stainless steel headbar was implanted on the skull. To do so, the skull was scraped clean, the headbar was lowered to touch the skull over lambda, then secured to the skull with a layer of dental adhesive (C and B Super-Bond, Sun medical) followed by dental cement (Jetkit, Lang). For fiber photometry experiments, a single fiber probe (200 μm, Chi Square Bioimaging or 200 μm, Doric Lenses, Quebec) was placed and fixed (C and B Super-Bond, Sun medical, UK) 150 μm above the injection site in the LHb. For all mice in this study, post-hoc analysis of the viral infection was performed after brain perfusion. Data were excluded when implant and injection were off-targeted.

## Viral constructs
The following constructs were used: rAAV5-GfaABC1D-GCaMP6f or rAAV9-GfaABC1D-GCaMP8s, rAAV5-hsyn-jRGECO1a (calcium imaging); rAAV9-hGFAP-Cre-mCherry; AAV5-hGFAP-EGFP-TVA-og and DG-Rabies-mCherry (monosynaptic retrograde labelling); rAAV1-hSyn1-mCherry; AAV5-gfaABC1D-mCherry (control); AAV9-hGFAP-hM3D(Gq)-mCherry for chemogenetic experiment. All constructs were purchased from Addgene or the Zurich University Vector Core, except the rabies construct, which was a gift from M. Schwartz from the University of Bonn.

## Fiber photometry recordings
Fiber photometry measurements were carried out with a ChiSquare X2-200 system (ChiSquare Biomaging, Brookline, MA) or Doric Lenses system. ChiSquare X2-200. Blue light from a 473–nm picosecond-pulsed laser (at 50 MHz; pulse width ∼80 ps FWHM) was delivered via a single mode fiber. Fluorescence emission from the tissue was collected by a multimode fiber with a sample frequency of 100 Hz. The single mode and multimode fibers were arranged side by side in a ferrule that is connected to a detachable multimode fiber implant. The emitted photons collected through the multimode fiber pass through a bandpass filter (FF01-550/88, Semrock) to a single-photon detector. Photons were recorded by the time-correlated single photon counting (TCSPC) module (SPC-130EM, Becker and Hickl, GmbH, Berlin, Germany) in the ChiSquare X2-200 system. Doric Lenses. Experiments were performed with one-site or two-site, two-color fiber photometry system measuring a 405 nm isosbestic and a 465 nm (GCaMP) or a 570 nm (jRGECO1a) on a single photodetector. Signals were recorded at 10 kHz using the built-in lock-in mode where 405 nm and 465 nm fiber-coupled LEDs focused into a 200 μm fiber coupled to the mouse optic fiber implant. Emitted light was collected through the same fiber, passed through an emission filter and detected by a photoreceiver module. The LED power was kept constant for every animal and every experimental session.

## Histology and immunofluorescence
Mice were terminally anesthetized with pentobarbital (150 mg.kg$^{-1}$) and perfused transcardially with 5–10 ml of 0.1 M phosphate buffered saline (PBS) and then paraformaldehyde (PFA 4%) in PBS. Brains were collected and left overnight in 4% PFA at 4 °C until slicing. Coronal slices (60 μm thick) of LHb were sectioned using a vibratome (Leica VT1200S). Slices were mounted on glass slides with FluorSave reagent. Brightfield and fluorescence images were acquired using an epifluorescence microscope (Zeiss).

For immunohistochemical analysis, brain sections were permeabilized at room temperature (RT) in 0.3% Triton X-100 (Sigma), followed by 1 h RT blocking in 5% Goat-Serum 0.5% Triton X-100 and overnight incubation with primary antibodies (1/500 NeuN, MAB377 Millipore, RRID AB_2298772 or 1/400 PCNA, Sigma-Aldrich, P8825, RRID AB_477413) at 4 °C. For Glutamine Synthetase (GS) immunostaining, the blocking solution was 15% Goat-Serum 2.5% Triton X-100, and the primary incubation was made for three overnights in 1.5% Goat-Serum 2.5% Triton X-100 solution (1/1000 Glutamine Synthetase, MAB302 Millipore, RRID AB_2110656) After washing, sections were incubated for 2 h RT with Alexa-fluorophore-conjugated secondary antibodies (Invitrogen). Confocal microscopy was performed with a Stellaris 5 (Leica) Laser Scanning System or a Stellaris 8 (Leica) Scanning System and images were processed and analyzed by FIJI/ImageJ Software. Confocal images for GCaMP6f/8 s and TVA-og signal, for NeuN, PCNA or GS immunofluorescence were used to manually outline positive cells.

## Experiments in Zebrafish
**Two-photon calcium imaging and sensory stimulation**. For in-vivo imaging, 3 weeks old juvenile zebrafish were embedded in 2.5% low-melting-point agarose (LMP, Fisher Scientific) in the lid of a 35 mm Petri dish. The constant perfusion of artificial fish water (AFW) bubbled with carbogen (95% O$_2$ and 5%CO$_2$) was maintained during the experiment. To ensure sufficient oxygen delivery to the animal, the LMP agarose was removed carefully in front of the nose, after solidifying for 20 min.

A two-photon microscope was used for calcium imaging: Scientifica Inc, with a 16x water immersion objective (Nikon, NA 0.8, LWD 3.0). For excitation, a mode-locked Ti:Sapphire laser (MaiTai Spectra-Physics) was

tuned to 920 nm. Recordings were performed as volumetric imaging (8 planes with a Piezo (Physik Instrumente (PI)). The acquisition rate was 2.33 Hz per plane (image size 1536×850 pixels).

First, spontaneous activity was measured for 12 min. Afterward, six repetitions of sensory stimuli (red light flash or vibration) were applied. For the light stimulus, we used a red LED (LZ1-00R105, LedEngin; 625-nm wavelength) and placed it in the front of the recording chamber near the tube. The light stimulus was a flash of 200 ms duration with an intensity of 0.318 mW. Vibrations were delivered via solenoid tapper (SparkFun Electronics, ROB-10391), via 200 ms application of 12 V. Total duration of the recordings was 30 min.

**Imaging of head-restrained zebrafish.** A Manta camera (at 120 Hz) was used to image the zebrafish. An IR light source (consisting of 780 nm LEDs, Thorlabs) was placed around the microscope objective straight above the fish to provide maximal IR illumination. The fish image was processed, and tail angle of the animal was recorded in real time.

**Zebrafish, statistics and reproducibility.** Active coping bouts were identified by thresholding for the tail angle (above 0.65 radian), and swim bout duration (above 1 second).

Two-photon microscopy images were aligned using suite2p. Recordings were then visually inspected for motion artifacts and Z-drift, recordings with remaining motion artifacts and Z-drift were discarded.

Habenular pixels corresponding to astroglia were identified through thresholding raw image. To facilitate analysis, we performed $10 \times 10$ binning of pixels. For each bin, fractional change in fluorescence ($\Delta F/F$) relative to baseline was calculated. We identified the motor-responsive bins by comparing the average activity during baseline (5 s-period before active coping bout) with the average activity during response window (10 s-period after the bout onset) using one-tailed Wilcoxon signed-rank test. Mean calcium signal of the identified bins was used in further analysis.

Statistical analysis was done using MATLAB; $p$-values are represented in the figure legends as ***$p < 0.001$. Wilcoxon signed rank test was used for the comparisons.

**Mice, statistics and reproducibility**

Animals were randomly assigned to experimental groups. All data are represented as mean ± SEM and individual data points are shown. Statistical testing was performed in GraphPad Prism 9.5.1 (GraphPad, San Diego, CA, USA) and Python using the Scipy package. Sample size was based on previous published work and experiments in the laboratory. The only exclusion criteria applied in the study was based on the correct injection of viral vector within the LHb. For behavioural analysis no exclusion criteria were applied. Animals were randomized at their arrival, but other confounders were not controlled for. All data was tested for normality using Shapiro-Wilk test. The experimenter was not blind of the experimental group. If normality was confirmed, paired or unpaired version of the two-tailed Student's t-tests was used for two-sample datasets, and one-way or one-way repeated measures ANOVA was used in case of more than two comparison groups. If data was not normally distributed, nonparametric tests were utilized instead: for two comparison groups, Wilcoxon signed-rank test and Mann-Whitney U-test for paired and unpaired testing respectively. In case of more than two comparison groups, Kruskal-Wallis test was used instead of ANOVA. For the experiment with DREADD–mediated LHb astrocyte activation, to account for the expected opposite direction of the effect in transition from Saline to CNO, compared to the transition from CNO to saline, change in total AC time and AC bout number from Saline to CNO was inverted before running one-way repeated measures ANOVA.

**Analysis of videotracking.** Active and Passive coping times were automatically scored using Python Scipy package as well as custom–written Python scripts (see data availability). In brief, motion index was extracted from behavioral videos using EthoVision software (CIT) based on the global pixel change in the arena. Resulting traces

were low-pass filtered using Butterworth first order filter with a cut-off frequency of 0.5 Hz. Active and passive coping bouts were auto-detected on filtered traces using amplitude- and duration-thresholding. Parameters of detection were chosen to fit the results of manual annotation on the test dataset, agnostic to the identities of experimental animals. Final thresholds chosen for the analysis of spontaneous AC events were 0.5 a.u. motion index for the amplitude, 2 s for minimum AC duration and 0.5 s for minimum PC duration. For the analysis of evoked AC events, threshold for minimum AC duration was increased to 2.7 s to filter out initial startle response of the animals. Due to some behavioral videos being acquired with a different (16.67 Hz instead of 10 Hz) framerate, to produce heatmaps of aligned AC events (Supplementary Fig. 1d, i, n, Supplementary Fig. 3p, r), Scipy function "resample" was used to interpolate values of higher frequency videos at 10 Hz.

**Behavioral experimental settings.** 4 weeks after viral injection and fiber implant, mice were recorded when being head–fixed and then exposed to aversive or appetitive stimuli or freely moving. Mice were habituated to the experimental room and then to a single-session (10 min) to the head fixation apparatus, using custom made headbars with screwpath, mice were fixed to the frame by fixing two screws. The next day, mice went through the head-fixed test (6 min). The following days mice were exposed to one behavioral challenge per day : 5 airpuffs (0.5 s, 0,5 bar) directed toward the eye, 5 aversive ultrasound (3 s tone from 17 to 20 kHz, 80 dB), 5 exposure to water-drop directed above the head, 5 exposure to an aversive blue light (0.5 s, 10 mW, 405 nm) or to 5 footshocks (0.3 mA, 0.5 s) given using a Ugo Basile shuttle box.

For appetitive stimuli, mice were tested in a social reward context, briefly mice were habituated to the arena with an empty enclosure for 10 min, a juvenile conspecific was then introduced in the enclosure and interactions were recorded. Regarding high-palatable food, mice were exposed to Nutella (Ferrero, Italy) in their homecage, then placed in an arena where a pipette tip was fixed to a wall. Their regular food pellets were placed on the other side of the cage, after 10 min habituation, around 2 g of Nutella was put on the pipette tip and licks were recorded.

The digging and grooming behaviors were recorded from the same mice when placed in an arena for 6 minutes.

For all behavioral challenges we z-scored each trial in reference to their baseline (3 s prior to the stimuli), for spontaneous AC, the reference baseline was 5 s, and the corresponding AC to PC baseline was used to z-score PC bouts. To compare AC and PC bouts, we calculated the average Z-score of the last second of AC or PC bout, in subsequent AC to PC transitions.

For the heatmap representation with all animals we normalized the data per animal, dividing all traces by the maximum z-score of the animal.

**Open field.** Mice in the open field were tracked using EthoVision software, coordinates of centroids were used to calculate velocity during the task. Velocity traces were low-pass filtered using Butterworth filter with 0.3 Hz frequency cut-off. To detect acceleration events we used custom Python scripts described above. Velocity threshold for the event was set at 6 cm/s, with minimum duration of 1 s and minimum preceding period of low velocity of 2 s.

To produce $Ca^{2+}$-velocity correlograms, photometry signal and velocity trace (motion index trace for head–restraint sessions) were synchronized and subsequently binned at 1 Hz.

**IP3R$^{fl/fl}$, iβark and hM3D(Gq) experiment.** 4 to 5 weeks after viral injection and headbar implant for IP3R$^{fl/fl}$ mice, iβark and hM3D(Gq) wild-type injected mice were recorded when being head–fixed for a period of 6 min. Regarding DREADD experiment, mice were habituated to intraperitoneous (IP) injections four times, one preceding the head-fixed habituation to the frame. On the experimental days, CNO or Saline was administered by IP injection, 30 minute before the head-fixed test.

**Analysis of Photometry.** Photometric signal traces were smoothed (constant time factor, 0.1 s) and further processed according to the trials using Spike2 software (Cambridge Electronic Design). When recorded with Doric System, motion artifacts in the GCaMP biosensor fluorescent channel (473 nm) or jRGECO1a (570 nm) were corrected by subtracting and then dividing by fluorescence values from isosbestic channel (405 nm).

**Analysis of monosynaptic connectivity with rabies strategies**. For anatomical tracing studies (C57BL6/6JR, Janvier, 10–15 week-old male and female mice were injected with rabies helper virus rAAV5-hGFAP-TVA-EGFP in the LHb (−1.5 mm AP, 0.45 mm ML, 2.6 mm DV from the skull surface). Animals were allowed to recover for three weeks, then re-injected at the same coordinates with modified glycoprotein deleted pseudotyped rabies-mCherry (RABV$\Delta$G (EnvA)-mCherry; titer: $1.10^7$pp/mL. Animals were transcardially perfused 7 days later and brains recovered. After sectioning and NeuN immunostaining, starting cells were counted in each animal, on 2 section level (200 μm away). Animals in which virus excessively spread outside LHb (>30%) or expressed in neurons (>0.5%) were excluded from analysis. Brain sections were analyzed in within the following antero-posterior coordinates: from 2.0 mm bregma to −4.0 mm bregma (roughly matching the prefrontal cortex and ventral tegmental area). The criteria for including a brain structure within the analysis was to exhibit >5 positive rabies cells on 2 adjacent sections in more than 3 animals. To ensure specificity of the TVA-og expression, sections were counterstained with an anti-GS antibody and cells inside LHb were counted. Furthermore, rabies positive cells in EPN, MHb and LH were also analyzed with the NeuN counterstaining to ensure their cell type.

### Reporting summary

Further information on research design is available in the Nature Portfolio Reporting Summary linked to this article.

## Data availability

The data sets generated during and/or analyzed during the current study are available online within the following Zenodo repository, https://doi.org/10.5281/zenodo.14843162[45].

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

## Acknowledgements

We thank members of the Mameli Laboratory as well as S. Valtcheva for discussions and comments on the manuscript. This work was supported by funds from the Canton of Vaud, the SNSF (310030_212193) and the State Secretariat for Education, Research and Innovation (SVEN) to M.M.; The Research Council of Norway FRIPRO grant 314212 and Centres of Excellence scheme, 332640 to E.Y. The authors would like to thank M. Schwarz at Bonn University for the kind gift of rabies constructs.

## Author contributions

L.M. and M.M. conceptualized the project and wrote the manuscript with the help of all authors. L.M. performed and analyzed all experiments with the help of D.O., S.L., P.M.; K.A.M. and E.Y. performed and analyzed zebrafish experiments. C.H., K.M., T.K. and A.V. provided the genetic mouse line as well as technical and conceptual support.

## Competing interests

The authors declare no competing interests.
