## [Transparent Peer Review file · Communications Biology]

Calcium dynamics in habenular astrocytes regulate active coping within behavioral transitions.

Corresponding Author: Professor Manuel Mameli

This manuscript has been previously submitted at another journal. This document only contains information relating to versions considered at Communications Biology.

Version 0:

Reviewer comments:

Reviewer #1

(Remarks to the Author)
Please see attached document

Reviewer #2

(Remarks to the Author)
In this manuscript, Michel et al. highlight a role of lateral habenula (LHb) astrocytes in state transitions. First, the authors established a behavioral challenge protocol whereby mice experienced active -to-passive behavioral state transitions. Using fiber photometry, they found that the calcium signals of LHb astrocytes were time-locked to active coping (AC) bouts followed by passive coping (PC). Moreover, they also monitored elevated Ca²⁺ signals of ventral habenula (vHb, LHb homologue) astrocytes in zebrafishes during AC to PC transitions. Disrupting astrocytic calcium signaling through IP3RKO of LHb astrocytes in mice prolongs active coping. Furthermore, using modified rabies-based viral strategies, the authors screened inputs of LHb astrocytes. Overall, I find the topic of broad interest, and the findings are novel and interesting.

Specific points:

1. The habenular neurons have been shown to exhibit progressive activation in behavioral state transitions (Andalman et al., 2019, Cell). In the current manuscript, the authors demonstrated calcium elevations during active-to-passive behavioral state transitions in habenular astrocytes. The temporal relationship of Ca²⁺ dynamics between habenular neurons and astrocytes remains unclear, and warrants further examination. Simultaneous monitoring of the activity dynamics of both cell types during behavioral state transitions should be tried.

2. The authors demonstrated that disrupting astrocytic calcium signaling prolongs active coping through IP3RKO of LHb astrocytes in IP3R-flox mice. It would be nice to verify this using an additional tool for astrocytic Ca²⁺ inhibition, e.g., iβARK, CalEx, which are technically feasible.

3. The authors employed modified rabies-based viral strategies to trace inputs onto LHb astrocytes. It is essential to quantitatively validate the efficiency and specificity of this viral tool. Also, high-resolution zoomed-in images of LHb starter cells in figure 2i and S4b should be provided.

4. Using the above tool, the authors made the intriguing observation that MHb neurons project to LHb astrocytes rather than neurons. What is the axonal organization of MHb-derived projection within the LHb subregion (e.g., medial/lateral; anterior/posterior parts)?

Reviewer #3

(Remarks to the Author)

The manuscript (COMMSBIO-25-1808-T) entitled “Calcium dynamics in habenular astrocytes regulate active coping within behavioral transitions” by Michel L. et al. provides evidence for the role of astrocyte, within the lateral habenula (LHb), in regulating transitions between active and passive coping in two animal models (mice and zebrafish). The application of novel rabies system for circuit tracing is also used to identify brain inputs to the astrocyte in the LHb.

Overall the study is well presented. However, there are several major and minor concerns that authors should address prior to being considered for publication;

- A major concern regards the lack of a solid control “status” to pin down the interpretation that the tracked motion in head-fixed mice is related to some kind of coping strategy. In head-fixed experiments, mice tend to move even after prolonged habituation and experimentation, and in the presence of both pleasant and aversive stimuli. Indeed, what is clear from the figures, and described in the text, is that the animal movement occurs spontaneously and is stable across time. There is a maximum of 50% chance of triggering movement upon various aversive triggers. So, when would this be active or passive coping? What happens if, instead of an air puff, mice receive sugar water or milk? I expect to see movement. So, when does the motion index indicate active coping, and when does it not? This aspect appears to be fundamental for interpreting the reported data and is currently not addressed.
- Another caveat is that mouse movement in head-fixed and in freely moving is very different, especially if a wheel or a treadmill is not used for the head-fixed setup. In this context, it becomes hard to know if the movement is an attempt of locomotion to escape from the situation (most likely) or not. Astrocyte calcium correlation to movement could be related to locomotion here as it was already observed in visual cortex.
- In the main text it is described that the forelimb and body movements were video-tracked but the method says that motion index was extracted using EthoVision. Since the extended video does not show tracked body parts or any other details, it remains unclear what represents the motion index (i.e., the overall change of pixels within the video, limb motion, or tail motion). This needs to be clarified.
- A standard behavioral test for depressive phenotype like AC/PC transition is learned helplessness in which mice slowly move from an active coping to a passive status. Is astrocyte calcium activity in LHb following this transition? Based on my previous comment, Fig. 2 indicates that astrocyte calcium activity correlates with mouse movement. This is something that has been shown in visual cortex as well and is linked to arousal (see publication from Kira Poskanzer’s lab), something that could explain the observed calcium activity in this manuscript as well. How does the calcium activity of astrocytes in the LHb change in a mouse model of depression?
- It is not clear the difference between Fig. 1g top and Ext.Fig. 2d bottom. Please clarify.
- In Ext. Fig. 3b please provide graphs for the number of NeuN and PCNA positive cells instead of the p-value. Staining for Sox9 or S100b would strengthen the analysis for the specificity of expression.
- The freely moving behavioral tests showed in Ext Fig. 3 f-i do not control for locomotion activity. A simple open field would fill the gap. Timestamps for the starting movement could be inferred using acceleration to extract references for the perievent graphs of calcium activity.
- Fig. 1d: the label of the virus is missing the mCherry in the hGFAP-Cre construct. It is confusing because not consistent with the text. Is the mCherry tethered to the Cre, or is a Cre-IRES-cherry? From the figure, I infer that it is the former case; please clarify.
- Fig. 1e, g: The axis labels would be more clear if IP3WT and IP3KO would be used.
- What is the explanation for the observed correlation between astrocyte calcium activity and movement (AC), if knocking out IP3R from astrocytes, so reducing calcium activity, results in an increased frequency of AC events?
- Most of the insets do not have a scale bar. Please fix this.
- The use of different model systems is a plus; it accounts for the understanding of conservative mechanisms governing neurobiological processes at a system level. However, the perievent raster plots and histogram for the PC bouts are missing for this data set, whereas they are shown for mice. Please add the graph in a supplementary figure.
- Is the neutral stimulus also eliciting tail movement? How can you control the mechanical stimulation given by the vibration device?
- A good example of learned helplessness (or giving up) behavior has been provided in zebrafish by using visual feedback for fish movement (see publication from Misha Ahrens` lab). Something similar would provide a behavioral control for AC/PC status.
- The use of the newly developed rabies is particularly informative for visualizing and understanding region-specific projections that contact astrocytes.
- Page 13: Correction needed. “Data were excluded when impant...” an “l” is missing.
- Page 17. Correction needed. “to produce heatmaps of aligned AC events (Figure 1f,l)” there is no “l” panel.

Version 1:

Reviewer comments:

Reviewer #1

(Remarks to the Author)

The authors have thoroughly addressed all my concerns raised during the initial review. The manuscript is significantly improved with the addition of new data, the incorporation of suggestions from all reviewers, and an expanded discussion. Overall, the findings are compelling, well supported by the presented data, and of clear interest to the field.

Reviewer #2

(Remarks to the Author)

The authors have satisfactorily addressed all the concerns raised in the previous review. The revisions are appropriate and improve the overall quality and clarity of the manuscript. I believe the study is now ready for publication.

Reviewer #3

(Remarks to the Author)

The revised manuscript (COMMSBIO-25-1808-T) entitled "Calcium dynamics in habenular astrocytes regulate active coping within behavioral transitions" by Michel L. et al. provides satisfying evidence to answer the major and minor points raised in my previous report. Hence, I thank all the authors for their effort and wish them good luck. The manuscript in my opinion is ready for publication.

Kind regards

Michel et al., Point by point to Reviewer comments

We would like to thank the reviewers for their constructive comments and the overall positive feedback on our work. We think the newly added experiments help to strengthen the conclusions of our study.

Reviewer #1

This manuscript examines the role of habenular astrocytes in active (AC) and passive (PC) coping behaviors. Using calcium imaging, genetic manipulation (IP3R deletion), and viral tracing in mice and zebrafish, the authors propose that habenular astrocytic calcium signaling regulates behavioral transitions. The study is novel and provides interesting insights into astrocytic contributions to behavior, but several methodological and conceptual concerns need to be addressed.

Major Concerns

1. 1. Justification for Studying Astrocytes and Citation Bias

- The rationale for studying astrocytes in this behavioral context is not well established. Prior research has demonstrated that astrocytes influence neuronal excitability, behavioral states, metabolic support, and synaptic plasticity. These aspects should be discussed to provide a broader context beyond the authors' own work.*
- The manuscript heavily references prior work from the authors' group (Volterra) but does not sufficiently acknowledge other contributions in the astrocytic field. Including key studies from groups working on astrocytic modulation of behavior would improve the balance of citations.*

We acknowledge the point raised by the reviewer. The initial submission was formatted as a Brief Communication which limited the total number of references. In the new submission, we extended the citations as suggested both within the introduction and discussion sessions. [See Line 88-89; 214-219]

2. Clarity in Experimental Protocol: The description of the head-fixed paradigm is extremely vague. What specifically makes this condition "inescapable"?

Clarify how it relates to established stress models. To me the rationale for habituation in a stress paradigm should be better justified, or the overall protocol better explain, not clear to me

As suggested by the reviewer, we have now extended the description of the protocol and experimental settings in the material and methods section. [See Line 687-690]

. Additionally, the video should be properly labelled in times and driving attention what the authors want to highlight.

We provide now a better illustrating video containing the movement tracking that is the motion index extracted for the plots. This should allow now the reader to identify the AC and PC bouts defined in the text.

• did authors measure how stress prior to introducing animals in the head-fixation could influence the bouts that alternated? by for example measuring them in an EPM?

The reviewer raises an interesting point. We agree that understanding how stress biases internal states and thus the AC-PC balance is a relevant question in the field. This research topic is ongoing in our laboratory and we think it deserves deep experimental work. We thus think it remains out of the scope of this work and more appropriate to more focused and detailed future studies.

3. Definition and Quantification of AC and PC: ' Altogether, these findings reveal that AC and behavioral transitions occur spontaneously, remain stable across time, and can be driven by external challenges according to their nature or intensities' according to the authors

•To make such a claim, the authors should compare the frequency of spontaneous AC between different conditions and the condition without triggered AC bouts. However, they are only comparing spontaneous and triggered AC, which is not sufficient.

We acknowledge the point raised by the reviewer. In the plot below we report the number of spontaneous AC bouts from all control conditions in the manuscript (HR session 1 post habituation). Next to this are plotted the spontaneous AC bouts in the sessions including the external challenges. To note that these experiments are done in subsequent sessions (daily). Thus, this indicates that AC bouts over sessions adapt although remaining stable in within sessions (Extended Data Figure 1). [One Way Anova, $df=3$, $F=5.55$]. [See Line 74-77]

- *The criteria for identifying AC vs. PC should be explicitly stated (e.g., movement thresholds, detection methods).*
- *The comparison between spontaneous and triggered AC bouts should be more precise.*

We apologize if the methods section was not exhaustive. We provide now further details related to thresholds, detection and analysis of all behaviors. [See Line 669-678]

4. 4. *Proving Sufficiency of Astrocyte Activity is Fundamental*

- *Optogenetic (melanopsin/ChR2) or chemogenetic activation of LHb astrocytes needs to be tested to determine if astrocyte activity alone can induce AC transitions. As should be the case if the claim of the authors state. Optionally, to be even more precise and what will massively improve the work is to show that real-time astrocyte manipulation could affect transitions à A closed-loop*

optogenetics approach could be used to test causality by activating or inhibiting astrocytes during AC-PC transitions.

We thank the reviewer for this suggestion. In a new set of experiments, we infused a viral vector in both Lhb enabling the expression of the excitatory dreadd (hM3D(Gq) to promote calcium increases in astrocytes after CNO and saline ip injections. In the new Extended Data Figure 2, we report that following alternated CNO/saline injections, AC bouts are modulated in opposing directions with respect to IP3RKO mice. [See Line 108-114]

Michel et al., Extended Data Fig. 2

5. Validation of Rabies Tracing: Rabies tracing in astrocytes, to my knowledge, has not been validated before. The paper referenced by the authors does not prove validation of this approach for astrocytes. Does the viral transmission follow the same logic as in neurons? The authors should confirm that TVA expression colocalizes with astrocyte markers (GFAP/S100) and is absent in neurons (NeuN) and further characterize this strategy.

We agree with the reviewer that this control is important. In a new set of experiments, we performed immunostaining against the GS (glutamine synthetase) and NeuN in cells expressing TVA. The quantification indicates that TVA expression is virtually limited to astrocytes. These data

are now presented in the new Figure 2 and Extended Data Figure 5. [See Line 175-178]

Michel et al., Extended Data Fig. 5

Minor Concerns

1. *Sentence Clarity: Some sections, especially in the methods, are difficult to follow and should be revised for clarity.*

Done.

2. *What is the reporter for the Cre-expressing astrocytes in Figure 2? It is clearly not mCherry, as the figure appears very different. However, this is not clearly indicated in the figure or its legend.*

Figure 2 presents few viral tools for GCaMP6/8 expression as well as TVA. In both cases the expression is driven by the astrocyte promoters (gfaABC1D or hGFAP). In this Figure there was no construct using Cre expression. In Figure 1d,e and Extended Data Figure 2a however, Cre expression is now clearly stating that mCherry was the reporter in the AAV-Cre. We apologize for the confusion in the initial submission.

3. *Discussion section should be expanded to include how Astrocyte-Neuron Communication might be occurring.*

- *The study assumes that astrocytes regulate LHb neurons but does not directly test gliotransmission. The authors could give some insight into what mechanisms should be further explored, such as blocking ATP/adenosine, glutamate, or D-serine to clarify this.*

- *Astrocytes regulate neuronal energy supply. Could metabolic support (e.g., lactate/glucose) contribute to AC-PC transitions?*

We extended the discussion around this point and provided literature in line with these topics. [See Line 217-224]

4. *Generalization Beyond Head-Fixation? What do the authors think the effects of astrocytes will be in freely moving paradigms (e.g., forced swim, tail suspension, EPM)? Would this confirm whether LHb astrocytes regulate stress responses more broadly?*

The reviewer raises an interesting point of view. To assess, at least partly this concern, in a new set of data we evaluated AC-driven astrocytes dynamics during a tail suspension test paradigm as a condition leading to active and passive coping. In the new Extended Data Figure 3, we report these findings where we observed large calcium transients in astrocytes time locked with vigorous movement bouts supporting the data obtained in head fixed mice. [See Line 127-129]

Michel et al., Extended Data Fig. 3

5. *Cross-Species Interpretation: The function of AC-PC transitions may differ between mice and zebrafish. Are these transitions stress responses in both species?*

This is an interesting point. Operationally, in both mice and zebrafish limiting mobility through head fixation leads to alternations between AC and PC. In both species this engages astrocytes dynamics. While both paradigms represent a challenge that constraint both mice and fish, commenting on similar stress responses might not be appropriate. Further work should address stress hormones elevations for instance in

both species or the intracellular pathways activated to better understand the point raised by the reviewer.

6. The manuscript alternates between “state transition” and “coping behavior.” Use consistent terminology.

We thank the reviewer for this note. We acknowledge this and we have made some adaptations in the text. However, we suggest keeping the term *state transition* when discussing AC to PC shifts and *coping behavior* when defining only one of this state. We hope the reviewer agrees with this strategy.

7. Photometry Data Processing: Explain how motion artifacts were corrected and clarify statistical treatment of signals.

The methods section has been now implemented with details related to dual wavelength and statistical details. [See Line 736-738]

8. Statistical Methods: More details on parametric assumptions and post-hoc corrections are needed.

Done. [See Line 650-663]

Reviewer #2

In this manuscript, Michel et al. highlight a role of lateral habenula (LHb) astrocytes in state transitions. First, the authors established a behavioral challenge protocol whereby mice experienced active -to-passive behavioral state transitions. Using fiber photometry, they found that the calcium signals of LHb astrocytes were time-locked to active coping (AC) bouts followed by passive coping (PC). Moreover, they also monitored elevated Ca²⁺ signals of ventral habenula (vHb, LHb homologue) astrocytes in zebrafishes during AC to PC transitions. Disrupting astrocytic calcium signaling through IP3RKO of LHb astrocytes in mice prolongs active coping. Furthermore, using modified rabies-based viral strategies, the authors screened inputs of LHb astrocytes. Overall, I find the topic of broad interest, and the findings are novel and interesting.

Specific points:

1. The habenular neurons have been shown to exhibit progressive activation in behavioral state transitions (Andalman et al., 2019, Cell). In the current manuscript, the authors demonstrated calcium elevations during active-to-passive behavioral state transitions in habenular astrocytes. The temporal relationship of Ca²⁺ dynamics between habenular neurons and astrocytes remains unclear, and warrants further examination. Simultaneous monitoring of the activity dynamics of both cell types during behavioral state transitions should be tried.

We thank the reviewer for this suggestion. In a new set of experiments, we expressed GCaMP8s in astrocytes and the red fluorescent calcium sensor jRGECO1a in neurons (New Extended Data Figure 4). We observed that neurons respond as well with an excitation at the onset of AC yet with less consistency and amplitude of transients. However, we do report short latencies responses in neurons than astrocytes which align with recently published data in the context of depression (Xin et al., 2025). This point is further expanded in the discussion. [See Line 147-155; 220-224]

Michel et al., Extended Data Fig. 4

2. The authors demonstrated that disrupting astrocytic calcium signaling prolongs active coping through IP3RKO of LHb astrocytes in IP3R-flox mice. It would be nice to verify this using an additional tool for astrocytic Ca²⁺ inhibition, e.g., β ARK, CalEx, which are technically feasible.

We thank the reviewer for this suggestion. In a new set of experiments,

we expressed β ARK (Nagai et al., 2021) to reduce the extent of calcium dynamics in astrocytes. In the new Extended Data Figure 2, we observed that this manipulation extended AC dynamics supporting a pivotal role of astrocytes in the transition between behavioral states similarly to what is observed in the IP3RKO. [See Line 104-108]

Michel et al., Extended Data Fig. 2

3. The authors employed modified rabies-based viral strategies to trace inputs onto Lhb astrocytes. It is essential to quantitatively validate the efficiency and specificity of this viral tool. Also, high-resolution zoomed-in images of Lhb starter cells in figure 2i and S4b should be provided.

In agreement with the reviewer, we have now used immunostaining against glutamine synthetase (GS) in cells expressing TVA. We find a large degree of co-localization between TVA and GS, but not with NeuN, supporting astrocyte specificity. This is now reported in the Extended Data Figure 5. As requested, we also provide the magnification images of starter cells. [See Line 175-177]

Michel et al., Extended Data Fig. 5

4. Using the above tool, the authors made the intriguing observation that MHB neurons project to LHb astrocytes rather than neurons. What is the axonal organization of MHB-derived projection within the LHb subregion (e.g., medial/lateral; anterior/ posterior parts)?

We thank the reviewer for raising this point. We did perform more analysis toward this objective. In our hands, axonal projections from the MHB remains quite medial in the LHb territory (New Extended Data Figure 5). This is corroborated with images obtained from the Allen Brain Atlas (see below). These data were obtained targeting Tac2, Tac1 and Chat neurons in MHB. We nevertheless think that this topic deserves a more detailed anatomical study and we would prefer providing here a qualitative proof of principle for the presence of axons. Future experimental effort will be needed to understand both the anatomical and the functional connectivity with astrocytes of the LHb.

Image source: Allen Mouse Brain Connectivity Atlas, Allen Institute for Brain Science.

Available at:

https://connectivity.brain-map.org/projection/experiment/-siv/265287564?imageId=265288266&imageType=TWO_PHOTON,SEGMENTATION&initImage=TWO_PHOTON&x=17317&y=11932&z=3

https://connectivity.brain-map.org/projection/experiment/-siv/268321927?imageId=268322117&imageType=TWO_PHOTON,SEGMENTATION&initImage=TWO_PHOTON&x=17161&y=10770&z=3

https://connectivity.brain-map.org/projection/experiment/-siv/300843826?imageId=300844113&imageType=TWO_PHOTON,SEGMENTATION&initImage=TWO_PHOTON&x=16129&y=9930&z=3

Reviewer #3

The manuscript (COMMSBIO-25-1808-T) entitled “Calcium dynamics in habenular astrocytes regulate active coping within behavioral transitions” by Michel L. et al. provides evidence for the role of astrocyte, within the lateral

habenula (LHb), in regulating transitions between active and passive coping in two animal models (mice and zebrafish). The application of novel rabies system for circuit tracing is also used to identify brain inputs to the astrocyte in the LHb.

Overall the study is well presented. However, there are several major and minor concerns that authors should address prior to being considered for publication;

1 A major concern regards the lack of a solid control “status” to pin down the interpretation that the tracked motion in head-fixed mice is related to some kind of coping strategy. In head-fixed experiments, mice tend to move even after prolonged habituation and experimentation, and in the presence of both pleasant and aversive stimuli. Indeed, what is clear from the figures, and described in the text, is that the animal movement occurs spontaneously and is stable across time. There is a maximum of 50% chance of triggering movement upon various aversive triggers. So, when would this be active or passive coping? What happens if, instead of an air puff, mice receive sugar water or milk? I expect to see movement. So, when does the motion index indicate active coping, and when does it not? This aspect appears to be fundamental for interpreting the reported data and is currently not addressed.

• Another caveat is that mouse movement in head-fixed and in freely moving is very different, especially if a wheel or a treadmill is not used for the head-fixed setup. In this context, it becomes hard to know if the movement is an attempt of locomotion to escape from the situation (most likely) or not. Astrocyte calcium correlation to movement could be related to locomotion here as it was already observed in visual cortex.

We thank the reviewer for raising this issue. We further extended our analysis to understand if astrocyte dynamics relate to movement. In the New Extended Data Figure 3, we analyzed velocity/acceleration in the open field and related this to the GCaMP8s signal and we used similar analysis for the head fixed. We observe that only in the context of AC in head fixation calcium transient are robust which is not the case during bouts of increased velocity corroborating our previous observation that

also during digging, reward intake or social interactions calcium dynamics were absent. [See Line 126-135]

Michel et al., Extended Data Fig. 3

2. In the main text it is described that the forelimb and body movements were video-tracked but the method says that motion index was extracted using EthoVision. Since the extended video does not show tracked body parts or any other details, it remains unclear what represents the motion index (i.e., the overall change of pixels within the video, limb motion, or tail motion). This needs to be clarified.

We apologize if this was not clear. In the method session we have now specified that the movement tracking was performed and detected based on pixel changes (Through Ethovision). We hope this clarifies the methods. Furthermore, we provided a better illustrating video containing the movement tracking that is the extracted motion index. [See Line 63-64; 667-669]

3. A standard behavioral test for depressive phenotype like AC/PC transition is learned helplessness in which mice slowly move from an active coping to a passive status. Is astrocyte calcium activity in LHb following this transition? Based on my previous comment, Fig. 2 indicates that astrocyte calcium activity correlates with mouse movement. This is something that has been shown in visual cortex as well and is linked to arousal (see publication from Kira Poskanzer's lab), something that could explain the observed calcium activity in

this manuscript as well. How does the calcium activity of astrocytes in the LHb change in a mouse model of depression?

We agree with the reviewer that this is an interesting aspect. However, this topic is better covered in recently published data from the laboratory of Hailan Hu. We nevertheless agree that understanding how stress biases internal states and thus the AC-PC balance is a relevant question in this context. This research topic is ongoing in our laboratory and we think it deserves deep experimental work in the future.

4. It is not clear the difference between Fig. 1g top and Ext.Fig. 2d bottom. Please clarify.

In Figure 1 the total AC duration during the session is reported, while in the Extended Data 2 the average duration per bout is reported. We made this more clear in the Figure legends.

5. In Ext. Fig. 3b please provide graphs for the number of NeuN and PCNA positive cells instead of the p-value. Staining for Sox9 or S100b would strengthen the analysis for the specificity of expression.

We agree with the reviewer that this is important. We provide now in the new Extended Data Figure 2 the quantification related to PCNA and GCaMP8s expression. Furthermore, in a new set of experiments, we performed immunostaining against glutamine synthetase (GS) in cells expressing TVA. These data are now presented in the Extended Data Figure 5. [See Line ; 175-177]

Michel et al., Extended Data Fig. 5

. The freely moving behavioral tests showed in Ext Fig. 3 f-i do not control for locomotion activity. A simple open field would fill the gap. Timestamps for the starting movement could be inferred using acceleration to extract references for the peri-event graphs of calcium activity.

As stated in point 1, in the New Extended Data Figure 3, we analyzed velocity in the open field and related this to the GCaMP8s signal and we used similar analysis for the head fixed task. The data support that calcium transient are elicited only in specific conditions but not during locomotion. [See Line 126-135]

7. Fig. 1d: the label of the virus is missing the mCherry in the hGFAP-Cre construct. It is confusing because not consistent with the text. Is the mCherry tethered to the Cre, or is a Cre-IRES-cherry? From the figure, I infer that it is the former case; please clarify.

We thank the reviewer for highlighting this inconsistency. Indeed, the Cre virus is tethered to an mCherry. This is now clarified in the figure and within the text. [See Line 100; 392-397]

8. Fig. 1e, g: The axis labels would be more clear if IP3WT and IP3KO would be used.

Done, this is now reported in the Figures.

9. What is the explanation for the observed correlation between astrocyte calcium activity and movement (AC), if knocking out IP3R from astrocytes, so reducing calcium activity, results in an increased frequency of AC events?

We thank the reviewer for sharing this thought. The correlation is obtained between each AC bout and the rise time of each calcium transient. Thus, for longer AC also the calcium transients are more spread in time. This scenario fits with the idea that calcium in astrocytes constrains AC, and indeed knocking down IP3R makes transients more rapid such that another transient can follow (as it will be not limited by the astrocyte function).

10. Most of the insets do not have a scale bar. Please fix this.

Done

11. The use of different model systems is a plus; it accounts for the understanding of conservative mechanisms governing neurobiological processes at a system level. However, the perievent raster plots and histogram for the PC bouts are missing for this data set, whereas they are shown for mice. Please add the graph in a supplementary figure.

Done, this is now included in the Extended Figure 4.

Michel et al., Extended Data Fig. 4

12. Is the neutral stimulus also eliciting tail movement? How can you control the mechanical stimulation given by the vibration device?

To have a better visualization of the point raised by the reviewer, below is a plot of the tail responses for both light (neutral) and vibration (aversive) stimuli (Onset set at 0s). The data suggest that the neutral stimulus does not elicit tail movement as the aversive stimulus. In addition, vibration strength is controlled by the voltage provided to the solenoid tapper.

13. *A good example of learned helplessness (or giving up) behavior has been provided in zebrafish by using visual feedback for fish movement (see publication from Misha Ahrens` lab). Something similar would provide a behavioral control for AC/PC status.*

We thank the reviewer for raising this point. Our work mostly addresses rapid responses to a challenge. The learned helplessness state measured by the laboratories of M Ahrens and K Deisseroth laboratories might represent “disrupted” AC-PC balance likely similar to a disease state. We acknowledge that this point is of interest but we think that to understand this relationship more experiments are necessary and feel this falls out of the scope of this work.

14. *The use of the newly developed rabies is particularly informative for visualizing and understanding region-specific projections that contact astrocytes.*

We thank the reviewer for this positive note.

15. *Page 13: Correction needed. “Data were excluded when impant...” an “l” is missing.*

Page 17. Correction needed. “to produce heatmaps of aligned AC events (Figure 1f,l)” there is no “l” panel.

Done

This manuscript examines the role of habenular astrocytes in active (AC) and passive (PC) coping behaviors. Using calcium imaging, genetic manipulation (IP3R deletion), and viral tracing in mice and zebrafish, the authors propose that habenular astrocytic calcium signaling regulates behavioral transitions. The study is novel and provides interesting insights into astrocytic contributions to behavior, but several methodological and conceptual concerns need to be addressed.

Major Concerns

1. Justification for Studying Astrocytes and Citation Bias

- The rationale for studying astrocytes in this behavioral context is not well established. Prior research has demonstrated that astrocytes influence neuronal excitability, behavioral states, metabolic support, and synaptic plasticity. These aspects should be discussed to provide a broader context beyond the authors' own work.
- The manuscript heavily references prior work from the authors' group (Volterra) but does not sufficiently acknowledge other contributions in the astrocytic field. Including key studies from groups working on astrocytic modulation of behavior would improve the balance of citations.

2. Clarity in Experimental Protocol: The description of the **head-fixed paradigm** is extremely vague. What specifically makes this condition "inescapable"? Clarify how it relates to established stress models. To me the rationale for **habituation in a stress paradigm** should be better justified, or the overall protocol better explain, not clear to me. Additionally, the video should be properly labelled in times and driving attention what the authors want to highlight.

- did authors measure how stress prior to introducing animals in the head-fixation could influence the bouts that alternated? by for example measuring them in an EPM?

3. Definition and Quantification of AC and PC: *'Altogether, these findings reveal that AC and behavioral transitions occur spontaneously, remain stable across time, and can be driven by external challenges according to their nature or intensities'* according to the authors

- To make such a claim, the authors should compare the frequency of spontaneous AC between different conditions and the condition without triggered AC bouts. However, they are only comparing spontaneous and triggered AC, which is not sufficient.
- The criteria for identifying **AC vs. PC** should be explicitly stated (e.g., movement thresholds, detection methods).
- The comparison between **spontaneous and triggered AC bouts** should be more precise.

4. Proving Sufficiency of Astrocyte Activity is Fundamental

- **Optogenetic (melanopsin/ChR2) or chemogenetic activation of LHb astrocytes** needs to be tested to determine if astrocyte activity alone can induce AC transitions. As should be the case if the claim of the authors state.

Optionally, to be even more precise and what will massively improve the work is to show that real-time astrocyte manipulation could affect transitions → A closed-loop optogenetics approach could be used to test causality by activating or inhibiting astrocytes during AC-PC transitions.

5. Validation of Rabies Tracing: Rabies tracing in astrocytes, to my knowledge, has not been validated before. The paper referenced by the authors does not prove validation of this approach

for astrocytes. Does the viral transmission follow the same logic as in neurons? The authors should confirm that TVA expression colocalizes with astrocyte markers (GFAP/S100) and is absent in neurons (NeuN) and further characterize this strategy.

Minor Concerns

1. **Sentence Clarity:** Some sections, especially in the methods, are difficult to follow and should be revised for clarity.

2. What is the reporter for the Cre-expressing astrocytes in Figure 2? It is clearly not mCherry, as the figure appears very different. However, this is not clearly indicated in the figure or its legend.

3. **Discussion section should be expanded to include how Astrocyte-Neuron Communication might be occurring.**

- The study assumes that astrocytes regulate LHb neurons but does not directly test gliotransmission. The authors could give some insight into what mechanisms should be further explored, such as blocking ATP/adenosine, glutamate, or D-serine to clarify this.

- Astrocytes regulate neuronal energy supply. Could metabolic support (e.g., lactate/glucose) contribute to AC-PC transitions?

4. **Generalization Beyond Head-Fixation?** What do the authors think the effects of astrocytes will be in freely moving paradigms (e.g., forced swim, tail suspension, EPM)? Would this confirm whether LHb astrocytes regulate stress responses more broadly?

5. **Cross-Species Interpretation:** The function of AC-PC transitions may differ between **mice and zebrafish**. Are these transitions stress responses in both species?

6. The manuscript alternates between “state transition” and “coping behavior.” Use consistent terminology.

7. **Photometry Data Processing:** Explain how **motion artifacts** were corrected and clarify statistical treatment of signals.

8. **Statistical Methods:** More details on parametric assumptions and post-hoc corrections are needed.

This study provides valuable insights into astrocytic regulation of behavioral transitions but requires significant **clarifications, and mechanistic validation**. Addressing these concerns will strengthen the manuscript’s impact and ensure robust conclusions.